# Quantitative localisation of titanium in the framework of titanium silicalite-1 using anomalous X-ray powder diffraction

Przemyslaw Rzepka [1,2,3,7], Matteo Signorile [4,7], Thomas Huthwelker [5], Stefano Checchia [6], Francesca Rosso [4], Silvia Bordiga [4] ✉ & Jeroen A. van Bokhoven [2,3] ✉

One of the biggest obstacles to developing better zeolite-based catalysts is the lack of methods for quantitatively locating light heteroatoms on the T-sites in zeolites. Titanium silicalite-1 (TS-1) is a Ti-bearing zeolite-type catalyst commonly used in partial oxidation reactions with $H_2O_2$, such as aromatic hydroxylation and olefin epoxidation. The reaction mechanism is controlled by the configuration of titanium sites replacing silicon in the zeolite framework, but these sites remain unknown, hindering a fundamental understanding of the reaction. This study quantitatively determines heteroatoms within the zeolite-type framework using anomalous X-ray powder diffraction (AXRD) and the changes in the titanium X-ray scattering factor near the Ti K-edge (4.96 keV). Two TS-1 samples, each with approximately 2 Ti atoms per unit cell, were examined. Half of the titanium atoms are primarily split between sites T3 and T9, with the remainder dispersed among various T-sites within both MFI-type frameworks. One structure showed significant non-framework titanium in the micropores of a more distorted lattice. In both samples, isolated titanium atoms were more prevalent than dinuclear species, which could only potentially arise at site T9, but with a significant energy penalty and were not detected.

Titanium silicalite-1 (TS-1) is a catalytically active zeolite-type material[1], widely used in several partial oxidation reactions[2–4]. Its unique abilities to epoxidize olefins with hydrogen peroxide ($H_2O_2$) and to release only water as a by-product are of clear interest from an industrial perspective, in particular concerning the epoxidation of propylene via the so-called HPPO (hydrogen peroxide to propylene oxide) process[5]. Owing to the outstanding properties in catalytically active processes, TS-1 is also one of the best-characterized materials. However, the mechanism that governs those processes still remains

unknown mostly because of the high dilution of titanium in the TS-1 siliceous framework (ca. 3 wt% $TiO_2$), hence hampering a precise assessment of the exact position of the metal within the possible crystallographic sites by conventional structural and spectroscopic techniques.

TS-1 structure belongs to **MFI**-type topology[6] and, like any zeolitic material, consists of tetrahedrally coordinated silicon atoms interconnected by covalently bonded oxygen bridges arranged into a porous construction. The complicated orthorhombic structure

[1]J. Heyrovsky Institute of Physical Chemistry Dolejškova 2155/3, 182 23 Prague 8, Czech Republic. [2]Institute for Chemical and Bioengineering, ETH Zurich, 8093 Zurich, Switzerland. [3]Paul Scherrer Institute, Center for Energy and Environmental Sciences, PSI, 5232 Villigen, Switzerland. [4]Department of Chemistry, NIS and INSTM Reference Centre, Università di Torino, Via G. Quarello 15, I-10135 and Via P. Giuria 7, I-10125 Torino, Italy. [5]Swiss Light Source, PSI, 5232 Villigen, Switzerland. [6]Beamline ID15A, European Synchrotron Radiation Facility 71 Avenue des Martyrs, 38000 Grenoble, France. [7]These authors contributed equally: Przemyslaw Rzepka, Matteo Signorile. ✉e-mail: silvia.bordiga@unito.it; jeroen.vanbokhoven@chem.ethz.ch

(*Pnma* space group) is constructed by two intersecting perpendicular systems of 10-membered ring channels (straight and sinusoidal) and 12 different crystallographic positions of silicon (T-sites). The outstanding catalytic performance of TS-1 in partial oxidation reactions presumably arises from the active sites that emerged by the isomorphous substitution of silicon for titanium. The minor concentration of tetrahedral titanium (up to ca. 3 wt% $TiO_2$) corresponds to around up to 2.5 Ti atoms per unit cell that can replace only a fraction of 8 symmetry-equivalent silicon positions at each of 12 T-sites (96 per unit cell in total). The fractional occupancy of T-sites and the intrinsic complexity of the TS-1 structure blurs the image of titanium distribution, although the material has been studied for years and by many different methods (NMR[7–9], other spectroscopic methods[10–14], and theory[15–19]). It remains unknown whether the active species responsible for the formation of Ti-peroxo-species are comprised of framework titanium monomers, di-titanium species, or higher nuclearity clusters and to what extent they reside within the crystalline framework[4,7]. Since the geometry of the sites is defined by the vicinity of the Ti positions in the framework, pinpointing titanium to specific T-sites is essential to understand their character. Despite the long-standing discussion, diffraction studies using synchrotron X-ray[20,21], and neutron radiation[22,23], also exploiting Ti isotope exchanged[24], have not managed to unambiguously elucidate the siting of titanium on the T-sites, partially due to its low concentration in the framework. Hence, to date, no definitive picture of the nature of the active site(s) in TS-1 has been established.

The primary objective of this study was the unequivocal identification of titanium siting within the TS-1 structure and determining whether titanium emerges as isolated species or dinuclear sites. Though X-ray atomic scattering factors $f = f_0 + f_1 + i \cdot f_2$ are primarily functions reflecting the number of electrons in elements ($f_0$) that should enable the discrimination of Ti (Z = 22) from Si (Z = 14) by using the conventional diffraction methods, the low concentration of Ti requires a technique that is more element sensitive, such as anomalous X-ray diffraction (AXRD) at the Ti K-edge. AXRD is a method that exploits the resonant scattering of atoms near their absorption edges[25,26]. Many, if not most, elements have absorption edges within

energy ranges suitable for anomalous diffraction experiments, e.g., Br (13.5 keV), Rb (15.2 keV), Se (12.66 keV), Zn (9.66 keV), and Fe (7.11 keV)[27,28]. Recently, aluminum was successfully pinpointed to the crystallographic T-sites in zeolite ferrierite[29], and cations on the extra-framework positions have been located[30–32]. Multiwavelength anomalous diffraction is broadly used to facilitate the structure determination of biological macromolecules[33].

## Results

### Ti anomalous X-ray scattering factors

To pinpoint titanium to T-sites in the TS-1 structure, the X-ray energy was set close to and far away from the Ti K-edge, near- and off-resonance. In the former case, the complex contributions of the scattering factor $f = f_0 + f_1 + i f_2$ become significant and the scattering power of the element changes dramatically while those from the other elements, silicon and oxygen, remain unchanged. Any changes to the pattern, therefore, relate to the location of titanium in the crystalline structure. Since the anomalous scattering on a coordinated titanium can differ from the one of the isolated atom as given in the LBL database[34], $f_1$ and $f_2$ values were derived from Ti K-edge XANES spectrum of TS-1 structure by mean of Kramers–Kronig relation (Fig. 1A)[35,36].

The different intensities of the Bragg reflections measured at off- and near-resonance energies vary and enable the most unequivocal assignment of the titanium to the T-atom in the framework structure. In contrast to short-range spectroscopic techniques, such as X-ray absorption spectroscopy, which probe only the local environments of Ti atoms, AXRD searches through long-range order and can quantify titanium at different T-sites and pinpoint their positions unambiguously in the crystal lattice.

Figure 1B depicts the variations in structure factors ($F_{hkl}$) for the TS-1 model, holding half of the Ti atom in the unit cell concentrated at the T3 site when energy is switched from off- (4.600 keV) to near-resonance (4.960 keV). The structural information on titanium distribution related to the specific lattice planes (hkl) can be quantified by the intensity change of the corresponding reflection ($I_{hkl}$) when the energy approaches the Ti K-edge. $F_{hkl}$ values are altered for the Ti-

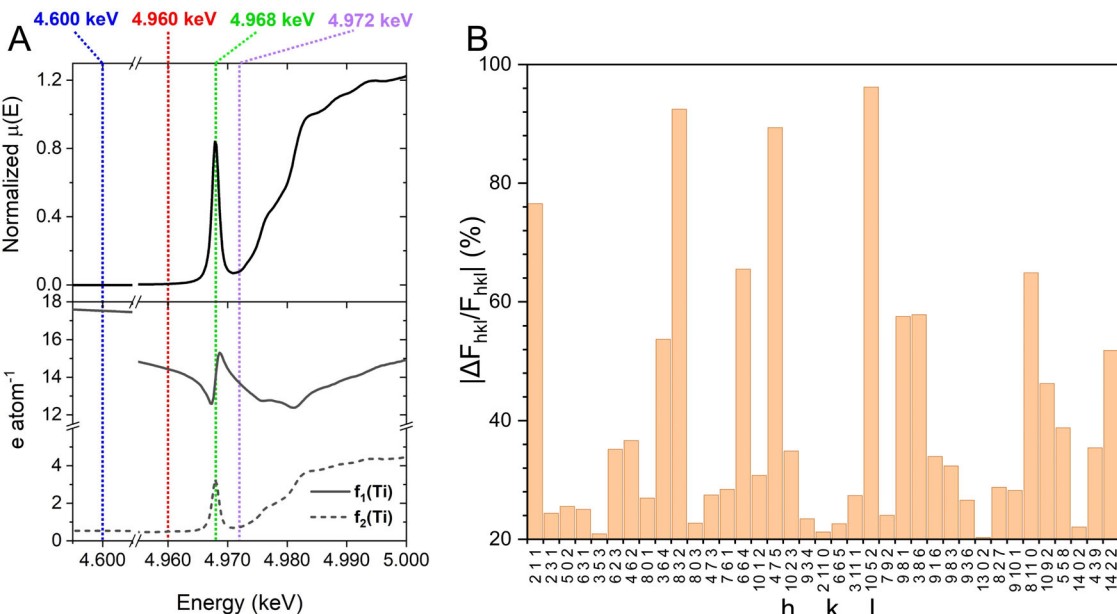

**Fig. 1 | Structure factors for TS-1 structure model. A** Top: Ti K-edge XANES recorded from TS-1A structure[13]. Bottom: The derived $f_1$ and $f_2$ contributions to titanium X-ray scattering factor near the Ti K-edge (4.96 keV). The diffraction data were collected at the near- and on-resonance energies represented by vertical lines. **B** $|\Delta F_{hkl}/F_{hkl}| = |(F_{hkl,4.600keV} - F_{hkl,4.960keV})/F_{hkl,4.600keV}|$: Relative differences between the selected structure factors ($\Delta F_{hkl} > 1$), $F_{hkl,E}$, calculated at off-resonance (E = 4.600 keV) and near-resonance (E = 4.960 keV) energies from a TS-1 structure model holding 0.5 Ti atoms per unit cell concentrated at T3 site.

bearing model, whereas those for Ti-free S-1 structure remain unchanged when changing the photon energy. $\Delta F_{hkl}$ values vary for different reflections and enable the titanium distribution to be determined.

### Location of titanium in TS-1 structures

Two industrial TS-1 samples (provided by Evonik Industries AG), originating from different synthetic routes and marked as TS-1A and TS-1B, were experimentally examined (basic characterization data available in Supplementary Fig. 1 and discussed therein). Both structures have similar chemical composition $Si_{96-x}Ti_xO_{192}$, where x ~ 2 (ca. 2.5 wt% $TiO_2$), and demonstrated similar catalytic activity in propylene epoxidation[14]. TS-1B contains some fraction of segregated $TiO_2$ (anatase polymorph) and hexacoordinated (i.e., distorted octahedral) Ti embedded in the **MFI** framework at Si-defect positions[13], that seem to slightly affect selectivity in HPPO reaction by increasing the concentration of reaction by-products (alcoholysis and hydrolysis products) at long reaction times[14].

The AXRD measurements were performed at the PHOENIX beamline at the Swiss Light Source synchrotron facility (SLS) operating at the tender energy ranges (https://www.psi.ch/en/sls/phoenix) using the previously developed set-up[29]. Since the 4.960 keV X-ray beam is totally absorbed after penetrating a few tens of micrometers of a zeolite sample, the experiment was carried out in reflection geometry inside the vacuum chamber. The preferred orientation of the zeolite crystallites in the vertical direction was prevented by the special sample preparation presented in the "Methods" section and depicted by SEM images (Supplementary Fig. 7). The silicon drift diode detector[37] enabled energy-specific detection of single photons. The discrimination of inelastic scattering suppressed the background elevated by the fluorescence emission released after an edge and enhanced the signal-to-noise ratio. Ti K-edge XANES scans were measured for each sample to determine the positions of the absorption edge. Further technical details regarding the experiments are summarized in the "Methods" section.

The analysis followed the protocol that was previously established[29]. The formulated data analysis strategy involved the single structure and profile parameters model refined with the Topas software[38] concurrently to fit all different energy diffraction datasets. The assignment of titanium to T-sites was preceded by the careful refinement of the orthorhombic TS-1 structure (*Pnma* space group) parameters against conventional diffraction data collected

in transmission mode (0.5 mm capillary) at wavelength $\lambda = 0.70860$ Å (E = 17.5 keV) on the Materials Science beamline at the SLS[39]. All diffraction data observed and those calculated from the refined models are shown in Supplementary Figs. 3, 4. There was no evidence of monoclinic distortions, as could be expected at low (<1 Ti per unit cell) loading[40]. A model that converged to the conventional diffraction data was simultaneously refined against the datasets collected across the Ti K-edge (Fig. 1A), at 4.600 keV (off-resonance), 4.960 keV, and 4.972 keV (both near-resonance) (Supplementary Figs. 3, 4). The dataset collected at 4.968 keV was excluded from the further evaluation since the Kramers–Kronig transformation of XANES data fails to decouple $f_1$ and $f_2$ values due to strong changes in the absorption coefficient for the on-resonance condition (Fig. 1A). At first, the profile parameters for AXRD data (scale factor, peak profile, specimen displacement, and axial model) were fitted. The initial scale factors were estimated using a previously established approach[29], which involved considering normalized $\mu_E$−attenuation coefficients for the sample composition at the specified energies and $I_0$−the incoming photon flux (calculated from the energy-dependent diode response function). Then the scales were allowed to be refined individually.

Once the profile functions had converged, the structure parameters were refined. The limit for observed diffraction at 4.960 keV corresponded to a $d$-spacing of 1.27 Å and enabled refining the atomic positions in the structure models against data across the absorption edge. Finally, the content of titanium, initially set as 2.5 Ti atoms equally divided between 12 T-sites, was refined while other parameters remained fixed at the values obtained before. The uncertainties of Ti occupancies on individual T-sites were determined by Bootstrap methods[41,42]. The details on the refinements (Supplementary Tables 1 and 2) and corresponding cif files are provided in the Supporting Information.

The refinement of TS-1A revealed 0.6(1) and 0.5(1) titanium per unit cell that occupy sites T3 and T9, respectively (Fig. 2A). This finding corroborates the theoretical calculations on this system that suggested those sites as the most stable titanium siting[15]. The tetrahedral titanium signal was also highlighted at other T-atoms of orthorhombic **MFI** framework[6], but the individual contributions at each of the sites remained insignificant. The refinement of total titanium content converged to 2.2 Ti atoms per unit cell in TS-1A. The tetrahedral Ti loading in the structure was reported to be 1.8[14]. Thus, all titanium is identified as a framework species.

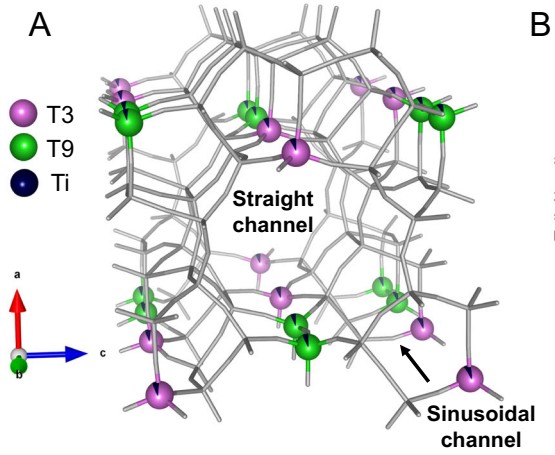
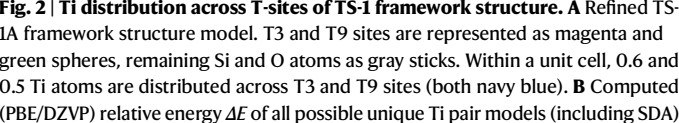
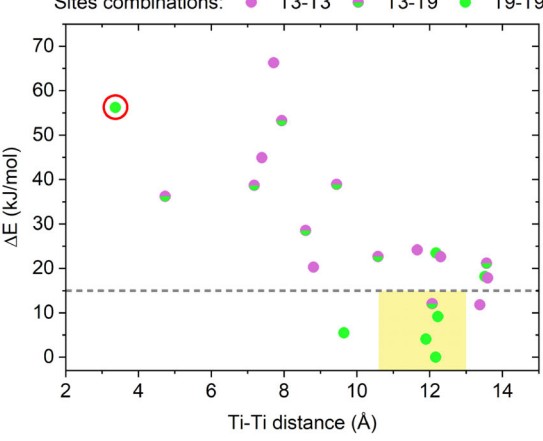

**Fig. 2 | Ti distribution across T-sites of TS-1 framework structure. A** Refined TS-1A framework structure model. T3 and T9 sites are represented as magenta and green spheres, remaining Si and O atoms as gray sticks. Within a unit cell, 0.6 and 0.5 Ti atoms are distributed across T3 and T9 sites (both navy blue). **B** Computed (PBE/DZVP) relative energy $\Delta E$ of all possible unique Ti pair models (including SDA) featuring the T3 and/or T9 site(s) as a function of the minimal Ti-Ti distance. The red circle points to the only model featuring the Ti-O-Ti moiety. The yellow box highlights the optimal range of Ti-Ti distances ($11.8 \pm 1.2$ Å), giving the most populated structures, identified as the average of the Ti-Ti distances typical of the six most stable ($\Delta E < 15$ kJ/mol) structures.

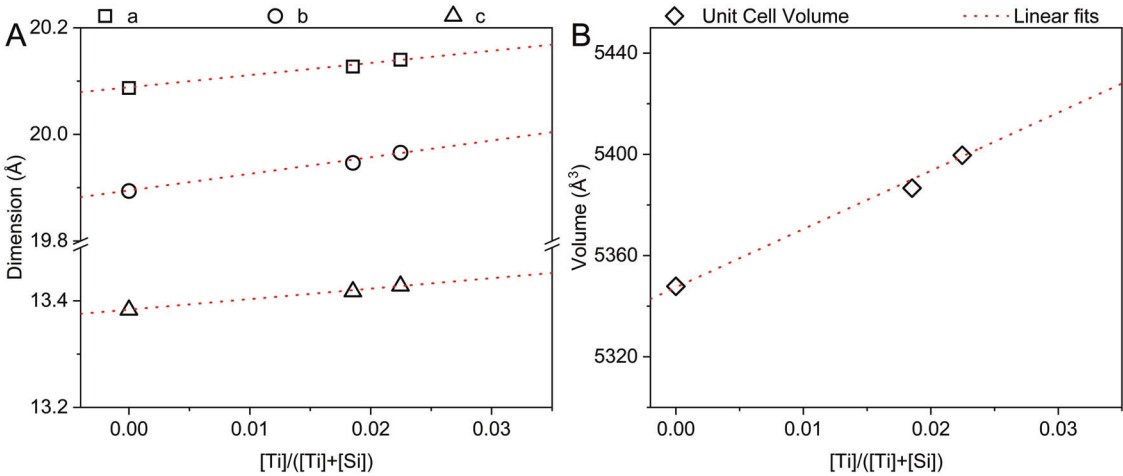

**Fig. 3 | TS-1 lattice expansion as a function of the Ti content.** Linear fits of **A** lattice parameters and **B** volume of the unit cell vs. the titanium content in TS-1 structures (*Pnma* space group). Ti-free S-1 was refined in monoclinic $P2_1/n11$ space group with $\alpha = 90.055(3)°$.

Whereas the TS-1B structure displays less distinct Bragg reflections than TS-1A (Supplementary Figs. 3, 4), it also shows a titanium population of 1.8 primarily hosted by sites T3 (0.5(1) Ti atom) and T9 (0.4(2) Ti atom), with the remaining titanium signal spread over various T-sites. Since the reported total Ti occurrence in the sample (2.89 wt% TiO$_2$)[14] would correspond to 2.1 Ti atoms per unit cell, the differential 0.3 titanium is non-framework. This stands in agreement with the previous findings on TS-1B that suggested over 10% contributions from higher coordination and non-framework titanium in the sample based on combined ICP, vibrational spectroscopy, and XAS studies[13,14].

The robustness of the refinement is further corroborated by the relations between the lattice parameters and Ti loadings in the samples as is shown in Fig. 3. The refined lattice parameters of orthorhombic TS-1B structure a = 20.127(5), b = 19.947(5), and c = 13.417(3) stay distinctly contracted in comparison to those revealed by TS-1A, respectively a = 20.140(2) Å, b = 19.966(2) Å, and c = 13.428(1) Å (Fig. 3). One can see the linear correlation between Ti/(Ti+Si) ratios in TS-1 and the lattice parameters when compared to those from Ti-free silicalite-1 (a = 20.089(3), b = 19.895(3), and c = 13.384(2)). Linear regression coefficients of unit cell expansion (Supplementary Table 3) corroborate with those reported by ref. 21 and ref. 40.

0.6(1) Ti atoms occupancy at a T3 site in the unit cell of TS-1A structure occurs as an isolated species that is equivalent to the T15 site in the lower monoclinic symmetry possible for the **MFI** framework, found as a probably occupied one[15]. On the other hand, the two nearest symmetry-equivalent positions ascribed to the T9 site, possibly occupied by titanium are located adjacent to each other resulting in four possible Ti pairs per unit cell (Fig. 2A). The oxygen bridge interconnecting those two titanium atoms points towards the intersection between straight and sinusoidal 10 ring channels and enables them to act as potential dinuclear titanium site, whose existence and reactivity in TS-1 is debated[7,13]. Differential Pair Distribution Function (dPDF) comparing the reference pattern of the defect-free Ti-free silicalite-1 (S-1) with G(r) of TS-1 structures (see SI for further details on sample preparation) provided no evidence supporting such preferential pairing (Supplementary Fig. 5), however, such measurement does not rule out that titanium dimers on T9 site emerge during synthesis and remain undetected due to extremely low occurrence of such ordering, additionally suppressed by the associated presence of silicon defect sites. Another possible structure with a quite short Ti-Ti distance may occur with a simultaneous occupation of a T3 and a T9 sites on the same five-membered ring and separated by a single siloxane bridge (i.e., generating a Ti-O-Si-O-Ti moiety). Though not properly a dimeric

Ti-O-Ti moiety, due to the metal centers' proximity, it may potentially act similarly under reaction conditions.

## Computational models of titanium pairs

To pinpoint the exact reciprocal position of T3 and/or T9 sites(s) within the unit cell, a set of periodic DFT calculations (PBE/DZVP, see Methods section for full computational details) were performed on the 22 possible unique pair models generated from the two sites by assuming a loading of two Ti atoms per unit cell (main computational results in Supplementary Table 4). Since the Ti distribution in the framework takes place during the hydrothermal crystallization of the zeolite framework, the structure directing agent (SDA), i.e., tetrapropylammonium hydroxide (TPA$^+$OH$^-$), has been explicitly included in the models. Previous attempts to explicitly consider the role of SDA in the simulation are rare and did not provide relevant support to the interpretation of experimental data[23]. The probability of having a given structure was evaluated on the basis of the relative energy of the different models. This is shown as a function of the minimal distance between the two Ti atoms in Fig. 2B. Known the uncertainty of DFT methods, also related to the selected computational setup[15,43], that can be in the order of 10–15 kJ/mol for the formation energy of siliceous zeolites[44], it clearly emerges that few structural models account for the larger fraction of sites (six models falls below a threshold of 15 kJ/mol). Furthermore, these structures are characterized by a very specific Ti-Ti distance, on average being 11.8 ± 1.2 Å. This site distribution significantly differs from that computed in the absence of the SDA, which shows much smaller differences in relative energies (16 models below the 15 kJ/mol threshold, see Supplementary Table 5), preventing a clear discrimination of siting.

## Discussion

A dimeric titanium model, capable of generating a bridging μ-η$^2$-peroxo species upon interaction with H$_2$O$_2$, was proposed on the basis of analogy of its NMR signatures with those of reference metalorganic compounds, and modeled by DFT considering a Ti-O-Ti moiety across the T7 and T11 positions (both located on the walls of the straight channel of **MFI** along the b-axis)[7]. The Ti-Ti distance for such a structure, evaluated by reoptimizing the model with the method adopted in this work (without including SDA), is 3.357 Å, comparable to the Ti-Ti distance for a Ti-O-Ti moiety where both Ti atoms sit on adjacent T9 sites (3.259 Å, in the absence of SDA as well). On the other hand, assuming the statistical distribution of 0.5(1) Ti atom across equivalent positions of the T9 site in the unit cell of TS-1 structure, there is only P = 1/28 probability to find two Ti atoms in the same unit cell (P = 1/4)

on two neighboring positions (P = 1/7). The probability further lowers considering the poor relative stability for this structure (56 kJ/mol over the most stable structure modeled by DFT), making the formation of dimeric Ti pairs at T9 sites unlikely on a thermodynamic basis. Thus, while dinuclear sites may theoretically emerge from bare geometrical considerations, their formation is accompanied by a severe energy penalty. The absence of distinctive features for titanium dimers, further corroborated by AXRD and dPDF and in combination with isolated siting at T3, suggests that monomeric titanium prevails in our materials.

In the mixed T3–T9 model, the Ti pair displays a Ti-Ti distance of 4.492 Å in absence of SDA, which appears too long to enable the formation of the $\mu_2$-$\eta^2$:$\eta^2$-peroxo species on geometrical basis (see also Supplementary Fig. 8). It could be possible that these titanium sites can still act together as a dimeric site, but hypothetically assuming a $\mu_2$-$\eta^1$:$\eta^1$-peroxo bridging coordination obtained through a different reaction mechanism. Nonetheless, such a Ti pair is also improbable on a thermodynamic basis due to its poor stability (36 kJ/mol) compared to the most probable models. Instead, structures with an "optimal" Ti-Ti distance (11.8 ± 1.2 Å) are much more stable. One can clearly see from Fig. 2B, as a general trend, that the relative stability of the considered models increases as Ti-Ti spacing increases. Within the six most stable structures, titanium will react with $H_2O_2$ as isolated Ti sites generating monomeric $\eta^2$-peroxo species, as broadly accepted in the literature[4]. The graphical representation for all commented models is provided in Supplementary Fig. 6.

In conclusion, TS-1A exclusively holds tetrahedral titanium, with more than half of it preferentially distributed across sites T3 and T9. The site T3 can host only isolated titanium species. Although dinuclear titanium sites could potentially emerge at adjacent T9 sites, such a model exhibits very low stability compared to more separated Ti sites and could only occur in the case of highly preferred localization experienced during sample synthesis. The framework titanium distribution in TS-1B resembles one in TS-1A, however ca. 15% titanium presence remains in non-framework locations. These two TS-1 catalysts of similar chemical composition $Si_{96-x}Ti_xO_{192}$ (x ~ 2) exhibit significant differences in the distribution of tetra- and octahedral titanium, despite both structures displaying elevated titanium concentration on similar T-sites within the MFI-type framework. The quantitative determination of heteroatoms in the TS-1 catalysts shows that titanium ordering in their structures follows an individual fashion of specific samples controlled by the catalysts' origins and can differ due to various synthesis conditions. Nonetheless, provided that Ti preferentially sits at namely isolated tetrahedral sites, TS-1 can be consistently described as a heterogeneous single-site catalyst, as broadly accepted in the literature[43]. Anomalous X-ray diffraction (AXRD) at the Ti K-edge constitutes the tool that allows to locate titanium of concentration even below 2.5 wt% $TiO_2$ in the TS-1 catalysts' structures of different origin.

## Methods

### The AXRD experiments at PHOENIX beamline

The experiments were performed on the PHOENIX (PHotons for the Exploration of Nature by Imaging and XAFS) undulator beamline at the Swiss Light Source (SLS) in Villigen, Switzerland that uses an elliptical undulator as the source. An in-vacuum diffractometer with Bragg-Brentano geometry was installed at the PHOENIX I branch that covers the energy range from 0.8–8 keV using a double crystal monochromator. A Si 111 crystal generated monochromatic light. High harmonics was rejected by reflections on three mirrors with a reflection angle of 0.4 degree. The diffractometer chamber, separated by a 500 nm thick silicon nitride window from the beamline, provided a residual pressure lower than $10^{-4}$ mbar by using a turbomolecular pump. The slit, made of simple razorblades, placed at a fixed distance, shaped the incoming beam into a horizontal line of 2 mm × 200 μm.

Two motorized goniometers rotated the sample holder and the detector to collect the diffracted photons in Θ-2Θ geometry. The sample holder was continuously rotating to compensate for the possible horizontal preferred orientation of the crystallites in the sample. The powder samples were dispersed in isopropanol and allowed to be set on a flat zero-diffraction Si wafer allowing all orientations of crystallites within a few micrometers of the sample layer. The specimen layer was also ensured to be settled flat to avoid micro-absorption effects on a rough surface. An energy-dispersive silicon drift diode (SDD, Viamp diode 50 mm², with MOXYEC AP3 window, manufacturer Ketek, Germany) set ~120 mm from the sample was used as the detector. Further details on the detector and the set-up exploited for AXRD experiments can be found in ref. 29.

To ensure the high statistical quality of all registered Bragg reflections, the weaker reflections were integrated for longer times to collect more photons compared to more intense reflections. All data points shown in the diffractograms are in arbitrary units proportional to the photon flux and normalized to the dwell time. The data are also corrected for the detector dead time.

### Computational details

DFT calculations were performed with the CP2K package (2022.1 version)[45,46] by employing the PBE GGA functional[47], including Grimme's D3 dispersion correction. Atomic cores were represented using Goedecker-Teter-Hutter (GTH) pseudopotentials[48]. The GPW method was adopted[49], being the Kohn-Sham orbitals expanded in the MOLOPT-DZVP basis set[50], while an auxiliary plane-wave basis (cutoff of 550 Ry) was used to represent the density. The energy in the SCF cycles was converged to $10^{-7}$ hartree, using the orbital transformation method[51]. Geometry optimization was conducted on atomic coordinates and cell parameters simultaneously, with the default convergence criteria (please reference to the CP2K online manual for further details: https://manual.cp2k.org/). Initially, a fully siliceous Silicalite-1 model including the $TPA^+OH^-$ SDA in the channels intersection (derived from the crystallographic structure by ref. 52) was optimized, then two Si occupying T3 and/or T9 sites were substituted by Ti. Only the 22 unique Ti pair models generated by substitution were considered, as obtained from the CONFCNT routine[53] implemented in the CRYSTAL17 code[54]. As Ti was inserted, an $OH^-$ anion from the nearest SDA molecule was manually coordinated to each Ti site, which is in line with the Parker and Millini hypothesis[8]. These structures were further optimized, and their relative energies $\Delta E$, with respect to the most stable optimized model, were computed as from Eq. (1):

$$\Delta E = E_i - E_{min} \tag{1}$$

where $E_i$ is the energy of the $i$th pair model, $E_{min}$ is the lowest energy among those of all possible Ti pair models. The main geometrical parameters, the relative energies, and the derived Boltzmann probabilities for the 22 pairs featuring two Ti at T3/T9 sites are given in Supplementary Table 4.

A set of models where the SDA was removed has been considered too. As the geometrical features of these structures (Supplementary Table 5) are obviously comparable with refinement data presented in this work, this second dataset was exploited for the sake of the computational method validation. The adopted method and models reproduced quite well the structural parameters experimentally found for TS-1. As an indication, the average RMSD for the calculated cell parameters with respect to the refined ones for TS-1B (having the closest Ti loading) is 0.13 Å. Also, the average Ti-O bond length is in line with previous literature (see ref. 15 and references herein). RESP charges for Ti atoms were computed for all models and their average (+1.88) was adopted to rescale the value of $f_1$ obtained via Kramers−Kronig transformation of XANES data.

## Pair distribution function

Total scattering data for X-ray pair distribution function (PDF) analysis was collected at ESRF (European Synchrotron Radiation Facility, Grenoble, France), on the ID15A beamline[55]. An X-ray wavelength of 0.2480 Å (50 keV) ensured contrast between Ti and Si (~5 barns/atom) while enabling scattering measurement up to a maximum scattering vector $Q_{max} = 22\,Å^{-1}$. Powders were contained in 0.7 mm diameter polyimide tubes and spun during measurement to ensure powder averaging. The scattering of air and empty polyimide tube in the same conditions was collected for subsequent background subtraction. The scattered radiation was collected by a Dectris Pilatus 2 M CdTe detector sitting at 206 mm from the sample. Raw images were azimuthally integrated, accounting for detector flat-field, spatial distortion, and beam polarization using pyFAI[56]. The resulting total scattering patterns were background-subtracted and converted to G(r) using PDFgetx3[57]. Fits to the experimental G(r) were done using Topas Academic v7 using the refined AXRD model as input structures.

## Data availability

Data supporting the findings of this study are available within the article and the Supplementary Information file. Source data are provided as a Source Data file. All raw data generated during the study can be obtained from P.R. and M.S. Source data are provided with this paper.

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

## Acknowledgements

P.R. thanks Dr. Ola Gjonnes Grendal from ID22 at ESRF for help with the Kramers–Kronig transformation of XANES data and Dr. Michal Andrzejewski from PSI for support with the ex situ capillary measurements. M.S., F.R., and S.B. acknowledge Evonik Resource Efficiency for providing samples investigated in this paper. M.S., F.R., and S.B. acknowledge Prof. Francesca Bonino and Prof. Valentina Crocellà for the fruitful discussion. M.S., F.R., and S.B. acknowledge support from the Project CH4.0 under the MUR program "Dipartimenti di Eccellenza 2023-2027" (CUP: D13C22003520001). M.S. thanks the C3S consortium for granting computational resources on the OCCAM cluster, funded by the Compagnia di San Paolo. P.R and J.A.v.B. acknowledge the Paul Scherrer Institute, Villigen, Switzerland, for the provision of synchrotron radiation beamtime on the PHOENIX and MS beamlines. S.C. acknowledges the ESRF for the provision of in-house research beamtime under proposal number IH-HC-4004 (10.15151/ESRF-ES-1555168862). All authors thank the reviewers, Dr. Roberto Millini and Prof. Ettore Fois, for their inputs, which were instrumental in enhancing the robustness and credibility of this publication.

## Author contributions

J.A.v.B. conceived the project, which was further developed by P.R., M.S., and S.B. P.R., M.S., and T.H. developed the methodology. P.R., M.S., T.H., S.C., and F.R. conducted the experiments. P.R., M.S., and S.C. performed the data analysis. All authors contributed to the data interpretation and writing the manuscript.

## Funding

## Competing interests

The authors declare no competing interests.
