## [Peer Review File · Nature Communications]

Quantitative locating titanium in the framework of titanium silicalite-1 by exploiting anomalous X-ray powder diffractionREVIEWERS' COMMENTS:

Reviewer #1 (Remarks to the Author):

The work, actually very well motivated and carried out, aims to answer one of the questions that those involved in heterogeneous catalysis with zeolites often ask themselves: are there preferential location sites for the heteroatoms, i.e. for catalytically active sites? This problem is particularly felt in the case of Titanium-Silicalite-1 (TS-1) a well-known catalyst for the oxidation of organic substrates, industrially applied in various processes, not least in the synthesis of propylene oxide. is a very complex problem, especially due to the very small amount of Ti atoms incorporated in the zeolitic framework (maximum 2.5 Ti atoms per unit cell composed of 96 tetrahedra [TO₄], with 12 crystallographically independent T sites in the orthorhombic structure). Several authors have tackled the problem both theoretically and experimentally, without however reaching consistent results (see Structure and Bonding, vol. 178 (2017), pp. 1 - 52). Here the authors exploit anomalous X-ray powder diffraction (AXRD) at the Ti K-edge, analyzing two different TS-1 samples provided by Evonik AG: one of them (TS-1A) contains Ti in tetrahedral coordination only (i.e. in framework position) while the other (TS-1B) clearly shows the presence of extra-framework species (anatase, amorphous phase, octahedrally coordinated Ti). The Authors reports that the analysis of the TS-1A sample led to the identification of the T3 and T9 sites as those most populated by Ti (0.6 and 0.6 Ti, respectively), while they consider insignificant the contribution of other sites where the Ti signal has been identified. For TS-1B sample the Authors argue that the situation is less clear: apart from site T3, ..."Ti occurrence is spread over various T sites".

After inspecting the data reported in the supplementary materials, I disagree with the Authors' view. I try to explain it with the following table, where for both samples the number of Ti atoms/u.c. for each site is reported, along with the relative percentage:

TS-1A TS-1B

Ti/u.c. % Ti/u.c. %

T1 0.32 14.3 0.00 0.0

T2 0.24 10.7 0.32 17.4

T3 0.56 25.0 0.48 26.1

T4 0.16 7.1 0.08 4.3

T5 0.00 0.0 0.00 0.0

T6 0.00 0.0 0.00 0.0

T7 0.16 7.1 0.00 0.0

T8 0.00 0.0 0.08 4.3

T9 0.48 21.4 0.40 21.7

T10 0.00 0.0 0.32 17.4

T11 0.16 7.1 0.00 0.0

T12 0.16 7.1 0.16 8.7

Total 2.24 1.84

Well, it is clear that also for the TS-1B sample the T3 and T9 sites are the most populated (0.5 and 0.4 Ti/u.c., respectively), exactly in the percentage detected for the TS-1B sample. Moreover, if we want to speculate on these numbers, even the statement that in the "TS-1B sample the Ti is spread over various sites" (i.e., that the distribution is more disordered) is not correct: in fact, considering the 4 most populated sites for each sample, it emerges that these contain 71.4% of the Ti in the TS-1A sample, and 82.6% in the TS-1B sample.

On the basis of these data, therefore, the discussion reported in the article as well as the conclusion that the distribution of Ti in the various sites depends on the synthesis procedure (by the way, not supported by data/references in the text) must certainly be revised.

Apart from this, I have many doubts about considering the results of the DFT calculations significant: in fact, these are conducted on the final model zeolite model, i.e. after calcination to eliminate the SDA (Tetrapropylammonium) cations present in the porous system, not taking into account that the incorporation of Ti into the framework takes place in the during the crystallization phase.

Let me be clear: as the experimental studies, I do not dispute the quality of the approach used,

rather than the selection/exclusion of the different configurations on the basis of even very small energy differences (a few kJ/mol), calculated on a structural model corresponding to the zeolite in the calcined form and not to that obtained from the synthesis (with the TPA present in the porous system)

Let me be clear: as the experimental studies, I do not dispute the quality of the approach used. Rather, I'm convinced that studying the distribution of Ti in the tetrahedral sites with the adopted model, basing the conclusions on the energy differences among the different configurations (of a few kJ/mol), is not correct because it is assumed that the structural model adopted (zeolite in the calcined form) corresponds to that present during the crystallization phase (with 4 TPA⁺/u.c.), and this that's not necessarily true.

For example, it is considered (and possibly refuted) the hypothesis formulated by Parker et al. in JACS 128 (2006) 1450 in which the Ti is five-coordinated and negatively charged to compensate for the positive charge of the TPA. If this hypothesis (never refuted) is correct, the incorporation of Ti and its localization would be strongly influenced by the interactions, even electrostatic, between the Ti site and the TPA cation. Furthermore, given the size of the TPA, it is at least difficult to hypothesize that there are configurations that envisage Ti sites such as the dinuclear ones (T9). In summary: I really appreciate the quality of the work from a methodological point of view, especially as regards the use of the AXRD technique in helping to resolve the long-standing question of the existence of preferential sites for heteroatoms, in particular, for Ti in the TS-1, a topic still being studied today, as demonstrated by the recent article by Gordon et al. who hypothesized the existence of dimeric Ti sites (ref. 12 in the ms).

However, I do not agree with the interpretation of the experimental evidence for the reasons specified above, just as it is my belief that the identification of the most probable configurations of Ti distribution in the unit cell can be conducted on the basis of relative energy data calculated through theoretical calculations on a structural model that does not correspond to the one actually existing during the phase of formation and crystallization of the zeolite.

Therefore, I recommend not accepting the paper in the present form, but I suggest a careful revision that takes into account the relevant aspects I have highlighted above

Reviewer #2 (Remarks to the Author):

In the manuscript entitled "Quantitative locating titanium in the framework of titanium silicalite-1 by exploiting anomalous X-ray powder diffraction at the Ti K-edge", the Authors present a methodology, based on AXRD, for the quantitative determination of heteroatoms location (Ti) in zeolite TS-1.

Although the adopted methodology (AXRD) has surely some potential in determining the precise location of heteroatoms in zeolite frameworks, I am not fully convinced that the presented data are sufficient for locating precisely and quantitatively Ti sites in TS-1.

As a matter of fact, in the Supplementary Information, where the Authors presented their x-ray data, I was not able to find convincing evidence for the Authors' claim of quantitative determination of Ti in TS-1. Indeed the reported crystallographic data show occupation uncertainty for Ti sites of the same order of the occupancy (some time larger). I think that these very large uncertainties do not allow one to draw final conclusions on the Ti siting in TS-1. The uncertainty issue should have been deeply discussed in the manuscript, however such a discussion is missing.

It should be noticed that, contrary to the Authors' claim "The absence of methods that enable to quantitatively locating heteroatoms .." (at line 13), there is at least one approach capable to determine interatomic distances between heteroatoms in solids (even amorphous) i.e. the Differential Pair Distribution Functions (DPDF) approach (see e.g. [dx.doi.org/10.1021/es200750b](https://doi.org/10.1021/es200750b) as an example). Also, see "Underneath the Bragg Peaks: Structural Analysis of Complex Materials" by T. Egami and S.J.L. Billinge (Pergamon). This approach should have been mentioned in the manuscript. Probably, a combination of DPDF and AXRD could be a winning strategy for the TS-1 case.

In Figure 1B, the Authors report simulated data on a model TS-1 enriched with 3 Ti atoms, all positioned in T9 sites. Such data are not significant for the case under investigation, because such a high Ti population on a single site of TS-1 is very unlikely, according to the literature (see e.g.

ref 15 of the manuscript). The Authors should have explained the rationale behind the choice of building a model with such a high occupancy on site T9.

A long discussion (page 9) concerns hypothetical reaction mechanisms catalyzed by TS-1.

However, it surprises me that the Authors use such hypothetical mechanisms to discriminate among different siting of Ti in TS-1. In principle, and given the manuscript title, it should be the other way round, namely from the experimentally determined Ti site occupancy one should be able to discriminate a particular mechanism.

In summary I do not support the publication of this manuscript on Nature Communications.

Reviewer #3 (Remarks to the Author):

The work by Rzepka and co-workers aims to locate titanium atoms in the frameworks of silicalite-1 zeolite based on anomalous X-ray powder diffraction data at the Ti K-edge. The topic is of highly interest as zeolites are among the most important heterogeneous catalysts at industrial level. Furthermore, small amounts of Ti incorporated into the framework can boost their performance in oxidation reactions.

The results are very interesting and open up a new approach for the location of heteroatoms in zeolitic framework; an aspect that is highly demanded in zeolitic chemistry due to their influence in the catalytic properties that could also potentially explain their stability. There are only few aspects that I think require further clarification.

From the abstract and from the manuscript, it seems that the authors established a method for the determination of heteroatoms; however, this methodology has already been employed in zeolites. Therefore, the method is not new and the novelty comes from the analysis of Ti within the framework. I believe this statement should be better clarified.

Again, in the abstract, authors mention that the "first structure showed the large contribution from non-framework titanium in the micropores and the disorder in the remaining framework titanium. Within a unit cell of the second structure, 0.6 and 0.5 titanium atoms were found concentrated at sites T3 and T9 of MFI-type framework. Pairing of two titanium atoms can only occur at site T9 in case of highly preferred localization experienced during sample production."

They finally conclude that TS-1A contains Ti at T3 and T9; while for TS-1B the Ti is more spread over different sites (besides the extraframework Ti). The abstract and the conclusions seem contradictory, or I misunderstood something?

Another aspect that I did not follow was: on page 4, figure 1B, the authors state that both TS-1 with Ti and free are depicted. However, only TS-1 with Ti is shown.

About sample preparation for the measurements, in the methods section the authors describe "The powder samples were dispersed in isopropanol and allowed to set on a 423 flat zero diffraction Si wafer allowing all orientations of crystallites within a few micrometers of the sample layer." Would it be possible to provide SEM data of the materials to actually see if different crystallite orientations were obtained? In fact, in the manuscript it is described that vertical direction was prevented by using this preparation method.

RESPONSE TO REVIEWERS' COMMENTS

Reviewer #1 (Remarks to the Author):

The work, actually very well motivated and carried out, aims to answer one of the questions that those involved in heterogeneous catalysis with zeolites often ask themselves: are there preferential location sites for the heteroatoms, i.e. for catalytically active sites? This problem is particularly felt in the case of Titanium-Silicalite-1 (TS-1) a well-known catalyst for the oxidation of organic substrates, industrially applied in various processes, not least in the synthesis of propylene oxide. is a very complex problem, especially due to the very small amount of Ti atoms incorporated in the zeolitic framework (maximum 2.5 Ti atoms per unit cell composed of 96 tetrahedra [TO4], with 12 crystallographically independent T sites in the orthorhombic structure). Several authors have tackled the problem both theoretically and experimentally, without however reaching consistent results (see Structure and Bonding, vol. 178 (2017), pp. 1 - 52).

Here the authors exploit anomalous X-ray powder diffraction (AXRD) at the Ti K-edge, analyzing two different TS-1 samples provided by Evonik AG: one of them (TS-1A) contains Ti in tetrahedral coordination only (i.e. in framework position) while the other (TS-1B) clearly shows the presence of extra-framework species (anatase, amorphous phase, octahedrally coordinated Ti).

The Authors reports that the analysis of the TS-1A sample led to the identification of the T3 and T9 sites as those most populated by Ti (0.6 and 0.6 Ti, respectively), while they consider insignificant the contribution of other sites where the Ti signal has been identified. For TS-1B sample the Authors argue that the situation is less clear: apart from site T3, ..."Ti occurrence is spread over various T sites".

After inspecting the data reported in the supplementary materials, I disagree with the Authors' view. I try to explain it with the following table, where for both samples the number of Ti atoms/u.c. for each site is reported, along with the relative percentage:

TS-1A	TS-1B			
Ti/u.c.	% Ti/u.c.	%		
T1	0.32	14.3	0.00	0.0
T2	0.24	10.7	0.32	17.4
T3	0.56	25.0	0.48	26.1
T4	0.16	7.1	0.08	4.3
T5	0.00	0.0	0.00	0.0
T6	0.00	0.0	0.00	0.0
T7	0.16	7.1	0.00	0.0
T8	0.00	0.0	0.08	4.3
T9	0.48	21.4	0.40	21.7
T10	0.00	0.0	0.32	17.4
T11	0.16	7.1	0.00	0.0
T12	0.16	7.1	0.16	8.7
Total	2.24	1.84		

Well, it is clear that also for the TS-1B sample the T3 and T9 sites are the most populated (0.5 and 0.4 Ti/u.c., respectively), exactly in the percentage detected for the TS-1B sample. Moreover, if we want to speculate on these numbers, even the statement that in the "TS-1B sample the Ti is spread over various sites" (i.e., that the distribution is more disordered) is not correct: in fact, considering the 4 most populated sites for each sample, it emerges that these contain 71.4% of the Ti in the TS-1A sample, and 82.6% in the TS-1B sample. On the basis of these data, therefore, the discussion reported in the article as well as the conclusion that the distribution of Ti in the various sites depends on the synthesis procedure (by the way, not supported by data/references in the text) must certainly be revised.

Answer:

We thank the reviewer for their support of the manuscript and emphasizing the need to identify heteroatoms within zeolite crystals. We also highly appreciate the reviewer for identifying and bringing an unfortunate oversight to our attention. We concur that in both structures, the distribution of tetrahedral titanium shows an elevated Ti concentration at sites T3 and T9. Therefore, the statement suggesting that 'the Ti is spread over various sites in TS-1B' is inaccurate. The observed differences between the samples stem from the contribution of non-framework titanium and distortions within the orthorhombic framework itself. Unfortunately, we are not allowed disclosing details on the synthetic procedure for these samples as such information is covered by a non-disclosure agreement with the company providing the materials.

As a result of this valuable feedback, we have made the necessary modifications to the following excerpts:

Line 26: Two representative TS-1 samples, each holding approximately 2 Ti atoms per unit cell, were examined. Half of the available titanium is primarily concentrated and distributed between sites T3 and T9 in both MFI-type frameworks, with the remaining signal dispersed among various T-sites. One of the samples exhibited a significant contribution from non-framework titanium in the micropores within the more distorted lattice. In either sample, isolated titanium atoms are notably more prevalent than dinuclear species. The latter could potentially arise only at site T9, which, however, incurs a significant energy penalty and was not detected.

Line 114: Two industrial TS-1 samples (provided by Evonik Industries AG), marked as TS-1A and TS-1B, were experimentally examined (basic characterization data available in Figure S1 and discussed therein).

Line 185: Whereas the TS-1B structure displays less distinct Bragg reflections than TS-1A (Figures S3 and S4), it also shows a titanium concentration of 1.8 primarily hosted by sites T3 (0.5(1) Ti atom) and T9 (0.4(2) Ti atom), with the remaining titanium signal spread over various T-sites. Since the reported total Ti occurrence in the sample (2.89 wt% TiO₂)¹⁴ would correspond to 2.1 Ti atoms per unit cell, the differential 0.3 titanium is non-framework.

Line 277: The framework titanium distribution in TS-1B resembles one in TS-1A, however ca. 15% titanium presence remains in non-framework locations. These two TS-1 catalysts of similar chemical composition $Si_{96-x}Ti_xO_{192}$ ($x \sim 2$) exhibit significant differences in the distribution of tetra- and octahedral titanium, despite both structures displaying elevated titanium concentration on similar T-sites within the MFI-type framework.

Apart from this, I have many doubts about considering the results of the DFT calculations significant: in fact, these are conducted on the final model zeolite model, i.e. after calcination to eliminate the SDA (Tetrapropylammonium) cations present in the porous system, not taking into account that the incorporation of Ti into the framework takes place during the crystallization phase.

Let me be clear: as the experimental studies, I do not dispute the quality of the approach used, rather that the selection/exclusion of the different configurations on the basis of even very small energy differences (a few kJ/mol), calculated on a structural model corresponding to the zeolite in the calcined form and not to that obtained from the synthesis (with the TPA present in the porous system)

Let me be clear: as the experimental studies, I do not dispute the quality of the approach used. Rather, I'm convinced that studying the distribution of Ti in the tetrahedral sites with the adopted model, basing the conclusions on the energy differences among the different configurations (of a few kJ/mol), is not correct because it is assumed that the structural model adopted (zeolite in the calcined form) corresponds to that present during the crystallization phase (with 4 TPA⁺/u.c.), and this that's not necessarily true.

For example, it is considered (and possibly refuted) the hypothesis formulated by Parker et al. in JACS 128 (2006) 1450 in which the Ti is five-coordinated and negatively charged to compensate for the positive charge of the TPA. If this hypothesis (never refuted) is correct, the incorporation of Ti and its localization would be strongly influenced by the interactions, even electrostatic, between the Ti site and the TPA cation.

Furthermore, given the size of the TPA, it is at least difficult to hypothesize that there are configurations that envisage Ti sites such as the dinuclear ones (T9).

Answer:

We acknowledge the Reviewer by their inspiring comment. As a matter of fact, though almost systematically neglected in computational siting studies, the structure-directing agent should be included in the simulation to account for more realistic synthetic conditions, i.e. when Ti is effectively being incorporated into the zeolite framework. Accordingly, we repeated our calculations by explicitly including TPA-OH in the model. The OH⁻ anion was coordinated to the nearest Ti site (in line with Parker hypothesis), with the TPA⁺ cation placed in the channels' intersection (as from crystallographic data on ZSM-5). As foreseen by the Reviewer, electrostatic contributions become dominant in determining the energy landscape and the variation among different site combinations becomes more evident in the order of tens of kJ/mol. Despite the large differences in energy compared to the previous version of the manuscript, we also reconsidered the

uncertainty associated to our periodic DFT approach. Systematic studies on the formation energies for elemental and simple (binary/ternary) compounds crystals revealed that an optimized DFT setup results in a mean average error of ca. 5 kJ/mol. Though studied in a less systematic way, the formation energies of zeolites are more challenging to compute accurately, with deviations that, on average, can account for 10-15 kJ/mol depending on the chosen functional/basis set combination. In a more conservative interpretation of the results, we considered only the relative energy of the different models considered in this work, as variations in the expected uncertainty could severely alter a site population based on the Boltzmann distribution. These results are outlined in Figure R1 (and, accordingly, in the revised Figure 2B, in Line 179):

Figure R1. Computed (PBE/DZVP) relative energy ΔE of all possible unique Ti pair models (including SDA) featuring the T3 and/or T9 site(s) as a function of the minimal Ti-Ti distance. The red circle points to the only model featuring the Ti-O-Ti moiety. The yellow box highlights the optimal range of Ti-Ti distances (11.8 ± 1.2 Å) giving the most populated structures, identified as the average of the Ti-Ti distances typical of the six most stable ($\Delta E < 15$ kJ/mol) structures.

Interestingly, by considering the effect of the SDA, the preferential distance among Ti sites increases to 11.8 Å (from 7.4 Å in the models without SDA), a value about half the cell parameters along a and b axes. This fact further supports the preference of Ti toward an “isolated” local environment, enforcing the broadly accepted model of isolated Ti sites as those active in partial oxidation reactions. We integrated the new findings in the version of the manuscript as follows:

Line 225: To pinpoint the exact reciprocal position of T3 and/or T9 sites(s) within the unit cell, a set of periodic DFT calculations (PBE/DZVP, see Methods section for full computational details) were performed on the 22 possible unique pair models generated from the two sites by assuming a loading of 2 Ti atoms per unit cell (main computational results in Table S4). Since the Ti distribution in the framework takes place during the hydrothermal crystallization of the zeolite framework, the structure directing agent (SDA), i.e. tetrapropylammonium hydroxide (TPA⁺OH⁻), has been explicitly included in the models. The probability of having a given structure was evaluated on the basis of the relative energy of the different models. This is shown as a function of the minimal distance between the two Ti atoms in Figure 2B. Known the uncertainty of DFT methods, in the order of 10-15 kJ/mol for the formation energy of siliceous zeolites,⁴³ it clearly emerges that few structure models account for the larger fraction of sites (6 models falls below a threshold of 15 kJ/mol). Furthermore, these structures are characterized by a very specific Ti-Ti distance, in average being 11.8 ± 1.2 Å. This sites distribution is significantly different compared to those computed in absence of the SDA, that reveals much smaller differences on the relative energies (16 models below the 15 kJ/mol threshold, see Table S5), that does not allow for a clear discrimination of siting.

Line 488: Initially, a fully siliceous Silicalite-1 model including the TPA⁺OH⁻ SDA in the channels intersection (derived from the crystallographic structure by van Koningsveld et al.⁵¹) was optimized, then 2 Si occupying T3 and/or T9 sites were substituted by Ti. Only the 22 unique Ti pair models generated by

substitution were considered, as obtained from the CONFCONT routine⁵² implemented in the CRYSTAL17 code.⁵³ As Ti was inserted, a OH⁻ anion from the nearest SDA molecule was manually coordinated to each Ti site, which is in line with the Parker and Millini hypothesis⁸. These structures were further optimized and their relative energies ΔE , with respect to the most stable optimized model, were computed as it follows:

$$\Delta E = E_i - E_{min}$$

where E_i is the energy of the i^{th} pair model, E_{min} is the lowest energy among those of all possible Ti pair models. The main geometrical parameters, the relative energies and the derived Boltzmann probabilities for the 22 pairs featuring 2 Ti at T3/T9 sites are given in Table S4.

In summary: I really appreciate the quality of the work from a methodological point of view, especially as regards the use of the AXRD technique in helping to resolve the long-standing question of the existence of preferential sites for heteroatoms, in particular, for Ti in the TS- 1, a topic still being studied today, as demonstrated by the recent article by Gordon et al. who hypothesized the existence of dimeric Ti sites (ref. 12 in the ms).

However, I do not agree with the interpretation of the experimental evidence for the reasons specified above, just as it is my belief that the identification of the most probable configurations of Ti distribution in the unit cell can be conducted on the basis of relative energy data calculated through theoretical calculations on a structural model that does not correspond to the one actually existing during the phase of formation and crystallization of the zeolite.

Therefore, I recommend not accepting the paper in the present form, but I suggest a careful revision that takes into account the relevant aspects I have highlighted above

Answer:

We sincerely thank the reviewer for the positive evaluation of the quality of our methodology. We acknowledge and agree with the reviewer's comment regarding the interpretation of the experimental evidence, for which we are greatly appreciative. We have reconsidered and have made the necessary corrections.

We believe that these changes have significantly improved the overall quality and accuracy of our work. The input was instrumental in enhancing the robustness and credibility of the data and we therefore acknowledge the reviewer in the acknowledgement section.

Reviewer #2 (Remarks to the Author):

In the manuscript entitled "Quantitative locating titanium in the framework of titanium silicalite-1 by exploiting anomalous X-ray powder diffraction at the Ti K-edge", the Authors present a methodology, based on AXRD, for the quantitative determination of heteroatoms location (Ti) in zeolite TS-1.

Although the adopted methodology (AXRD) has surely some potential in determining the precise location of heteroatoms in zeolite frameworks, I am not fully convinced that the presented data are sufficient for locating precisely and quantitatively Ti sites in TS-1.

As a matter of fact, in the Supplementary Information, where the Authors presented their x-ray data, I was not able to find convincing evidence for the Authors' claim of quantitative determination of Ti in TS-1. Indeed the reported crystallographic data show occupation uncertainty for Ti sites of the same order of the occupancy (some time larger). I think that these very large uncertainties do not allow one to draw final conclusions on the Ti siting in TS-1. The uncertainty issue should have been deeply discussed in the manuscript, however such a discussion is missing.

Answer:

We appreciate the thorough examination of our manuscript by the reviewer and the opportunity to address their concerns. In response to the reviewer's comment, we clarify some key points.

Firstly, we acknowledge the concerns about the reported uncertainties in the occupancy of Ti sites within zeolite TS-1. We emphasize that the uncertainties were evaluated using the Singular Value Decomposition method as implemented in Topas Academic.¹ This method, while commonly used, has specific limitations, especially when dealing with site occupancies in complex systems.^{2,3} To address this issue, we have now adopted the bootstrap_errors algorithm, which is recognized for its versatility and robustness in estimating

uncertainties in crystallography, including site occupancies. This approach can better account for non-linear relationships, constraints, and complex dependencies between parameters, and we believe it will provide a more accurate representation of the uncertainties. The following line was added:

Line 161: The uncertainties of Ti occupancies on individual T-sites were determined by Bootstrap methods.^{41,42}

Secondly, the concentrations of titanium at sites T3 and T9 are, in fact, more than three times higher than the estimated uncertainties (3σ). The signal at the remaining sites was not discussed, as it remained within the noise. We are thus confident about the enrichment of titanium on specific T-sites.

- (1) Coelho, A. A. TOPAS and TOPAS-Academic: An optimization program integrating computer algebra and crystallographic objects written in C++. *J. Appl. Crystallogr.* 51, 210–218 (2018).
- (2) Efron, B. & Tibshirani, R. Bootstrap methods for standard errors, confidence intervals, and other measures of statistical accuracy. *Stat. Sci.* 1, 54–75 (1986).
- (3) DiCiccio, T. J. & Efron, B. Bootstrap confidence intervals. *Stat. Sci.* 11, 189–212 (1996).

It should be noticed that, contrary to the Authors' claim "The absence of methods that enable to quantitatively locating heteroatoms .." (at line 13), there is at least one approach capable to determine interatomic distances between heteroatoms in solids (even amorphous) i.e. the Differential Pair Distribution Functions (DPDF) approach (see e.g. dx.doi.org/10.1021/es200750b as an example). Also, see "Underneath the Bragg Peaks: Structural Analysis of Complex Materials" by T. Egami and S.J.L. Billinge (Pergamon). This approach should have been mentioned in the manuscript. Probably, a combination of DPDF and AXRD could be a winning strategy for the TS-1 case.

Answer:

We thank the reviewer for this insightful remark that made us recognize the potential of the dPDF approach and fully appreciate its value in determining interatomic distances between heteroatoms. Despite the challenges posed by a diffusive signal from titanium diluted by the presence of silicon defect sites in titanium silicalite-1, we successfully conducted the respective analysis. The results obtained corroborate and support the findings from AXRD and DFT calculations. We have meticulously presented and discussed these results in the Supplementary Information:

Line 224: **Pair Distribution Function**

The radially-averaged distribution of distances between titanium atoms that belong to T3 and/or T9 sites(s) has been analyzed by the pair distribution function (PDF) method based on 50keV total scattering data (Figure S4A). The $G(r)$ function of each sample was correctly reproduced by the respective structural model derived from AXRD, after relaxing atomic positions to account for correlated atomic displacements and peak broadening. The $G(r)$ observed for the reference defect-free Ti-free silicalite-1 (S-1) was subtracted from the $G(r)$ for TS1A and TS-1B (Figure S4B). The resulting Differential Pair Distribution Function (dPDF) highlighted the signal from a diluted dopant by excluding any interatomic distances shared by all samples.¹⁰ The real-space distance 1.84 Å corresponds clearly to Ti-O bond length and negative peaks 1.62 Å and 3.11 Å indicate the deficiency of Si-O and Si-O-Si pairs respectively. This confirms that titanium substitutes silicon as a heteroatom in the TS-1 structure. The remaining peaks either overlap with those generated by $\Delta G(r)$ calculated by subtracting the reference S1 from the defected silicalite-1 (DS-1), hence can be assigned to the defect sites, or result from peak broadening of $G(r)$ from both TS-1, which is associated with local disorder in the structures.

Figure S5. (A) Pair distribution functions of titanium silicalite-1 (TS-1A and TS-1B), titanium-free silicalite-1 (S-1), defected silicalite-1 (DS-1), and refined *P1* models. (B) Differential pair distribution function generated by subtracting the PDF of S-1 from those of TS-1A, TS-1B, and DS-1.

The approach taken is described in the Methods section:

Line 512: **Pair Distribution Function**

Total scattering data for X-ray Pair Distribution Function (PDF) analysis was collected at ESRF (European Synchrotron Radiation Facility, Grenoble, France), on the ID15A beamline.⁵⁴ An X-ray wavelength of 0.248 Å (50 keV) ensured contrast between Ti and Si (~5 barns/atom) while enabling scattering measurement up to a maximum scattering vector $Q_{\text{max}}=22 \text{ \AA}^{-1}$. Powders were contained in 0.7 mm diameter polyimide tubes and spun during measurement to ensure powder averaging. The scattering of air and empty polyimide tube in the same conditions was collected for subsequent background subtraction. The scattered radiation was collected by a Dectris Pilatus 2M CdTe detector sitting at 206 mm from the sample. Raw images were azimuthally integrated accounting for detector flat-field, spatial distortion, beam polarization using pyFAI.⁵⁵ The resulting total scattering patterns were background-subtracted and converted to $G(r)$ using PDFgetx3.⁵⁶ Fits to the experimental $G(r)$ were done using Topas Academic v7 using the refined AXRD model as input structures.

In Figure 1B, the Authors report simulated data on a model TS-1 enriched with 3 Ti atoms, all positioned in T9 sites. Such data are not significant for the case under investigation, because such a high Ti population on a single site of TS-1 is very unlikely, according to the literature (see e.g. ref 15 of the manuscript). The Authors should have explained the rationale behind the choice of building a model with such a high occupancy on site T9.

Answer:

The figure 1B was replaced by R2 that demonstrate the difference in structure factors for a model with 0.5 Ti atom only. The differences remain significant.

Figure R2. $|\Delta F_{hkl}/F_{hkl}| = |(F_{hkl,4.600\text{keV}} - F_{hkl,4.960\text{keV}})/F_{hkl,4.600\text{keV}}|$: Relative differences between the selected structure factors ($\Delta F_{hkl} > 1$), $F_{hkl,E}$, calculated at off-resonance ($E = 4.600$ keV) and near-resonance ($E = 4.960$ keV) energies from a TS-1 structure model holding 0.5 Ti atoms per unit cell concentrated at T3 site.

A long discussion (page 9) concerns hypothetical reaction mechanisms catalyzed by TS-1. However, it surprises me that the Authors use such hypothetical mechanisms to discriminate among different siting of Ti in TS-1. In principle, and given the manuscript title, it should be the other way round, namely from the experimentally determined Ti site occupancy one should be able to discriminate a particular mechanism.

Answer:

We acknowledge the Reviewer for the useful comment. We agree that, considered the scope of this manuscript, the previous formulation of the structure vs. reactivity discussion must be improved. We rephrased the whole paragraph by first presenting the structural evidences from our study, then putting these in connection with the broader discussion on potential active sites/reaction mechanisms:

Line 206: 0.6(1) Ti atoms occupancy at a T3 site in unit cell of TS-1A structure occurs as an isolated species that is equivalent to the T15 site in the lower monoclinic symmetry possible for the **MFI** framework, found as a probably occupied one.¹⁵ On the other hand, the two nearest symmetry-equivalent positions ascribed to the T9 site, possibly occupied by titanium are located adjacent to each other resulting in 4 possible Ti pairs per unit cell (Figure 2A). The oxygen bridge interconnecting those two titanium atoms points towards the intersection between straight and sinusoidal 10 ring channels and enable them to act as potential dinuclear titanium site, whose existence and reactivity in TS-1 is debated.^{7,13} Differential Pair Distribution Function (dPDF) comparing the reference pattern of the defect-free Ti-free silicalite-1 (S-1) with $G(r)$ of TS-1 structures (see SI for further details on sample preparation) provided no evidence supporting such preferential pairing (Figure S5), however such measurement does not rule out that titanium dimers on T9 site emerge during synthesis and remain undetected due to extremely low occurrence of such ordering additionally suppressed by associated presence of silicon defect sites. Another possible structure with quite short Ti-Ti distance may occur with a simultaneous occupation of a T3 and a T9 sites on the same 5-membered ring and separated by a single siloxane bridge (i.e. generating a Ti-O-Si-O-Ti moiety). Though not properly a dimeric Ti-O-Ti moiety, due to the metal centres proximity, it may potentially act similarly under reaction conditions.

To pinpoint the exact reciprocal position of T3 and/or T9 sites(s) within the unit cell, a set of periodic DFT calculations (PBE/DZVP, see Methods section for full computational details) were performed on the 22 possible unique pair models generated from the two sites by assuming a loading of 2 Ti atoms per unit cell (main computational results in Table S4). Since the Ti distribution in the framework takes place during the hydrothermal crystallization of the zeolite framework, the structure directing agent (SDA), i.e. tetrapropylammonium hydroxide (TPA⁺OH⁻), has been explicitly included in the models. The probability of having a given structure was evaluated on the basis of the relative energy of the different models. This is

shown as a function of the minimal distance between the two Ti atoms in Figure 2B. Known the uncertainty of DFT methods, in the order of 10-15 kJ/mol for the formation energy of siliceous zeolites,⁴³ it clearly emerges that few structure models account for the larger fraction of sites (6 models falls below a threshold of 15 kJ/mol). Furthermore, these structures are characterized by a very specific Ti-Ti distance, in average being 11.8 ± 1.2 Å. This sites distribution is significantly different compared to those computed in absence of the SDA, that reveals much smaller differences on the relative energies (16 models below the 15 kJ/mol threshold, see Table S5), that does not allow for a clear discrimination of siting.

A dimeric titanium model, capable of generating a bridging μ - η^2 -peroxo species upon interaction with H_2O_2 , was proposed on the basis of analogy of its NMR signatures with those of reference metalorganic compounds, and modelled by DFT considering a Ti-O-Ti moiety across the T7 and T11 positions (both located on the walls of the straight channel of **MFI** along the b-axis).⁷ The Ti-Ti distance for such structure, evaluated by reoptimizing the model with the method adopted in this work (without including SDA), is 3.357 Å, comparable to the Ti-Ti distance for a Ti-O-Ti moiety where both Ti atoms sit on adjacent T9 sites (3.259 Å, in absence of SDA as well). On the other hand, assuming the statistical distribution of 0.5(1) Ti atom across equivalent positions of T9 site in unit cell of TS-1 structure, there is only $P=1/28$ probability to find two Ti atoms in the same unit cell ($P=1/4$) on two neighbouring positions ($P=1/7$). The probability further lowers considering the poor relative stability for this structure (56 kJ/mol over the most stable structure modelled by DFT), making the formation of dimeric Ti pairs at T9 sites unlikely on a thermodynamic basis. Thus, while dinuclear sites may theoretically emerge from bare geometrical considerations, their formation is accompanied by a severe energy penalty. The absence of distinctive features for titanium dimers further corroborated by AXRD and dPDF and in combination with isolated siting at T3, suggests that monomeric titanium prevails in our materials.

In the mixed T3-T9 model, the Ti pair displays a Ti-Ti distance of 4.492 Å in absence of SDA, which appears too long to enable the formation of the μ - η^2 -peroxo species on geometrical basis. It could be possible that these titanium sites can still act together as a dimeric site, but hypothetically assuming a μ - η^1 -peroxo bridging coordination obtained through a different reaction mechanism. Nonetheless, such a Ti pair is also improbable on a thermodynamic basis due to its poor stability (36 kJ/mol) compared to the most probable models. Instead, structures with an "optimal" Ti-Ti distance (11.8 ± 1.2 Å) are much more stable. One can clearly see from Figure 2B, as a general trend, that the relative stability of the considered models increases as Ti-Ti spacing increases. Within the six most stable structures, titanium will react with H_2O_2 as isolated Ti sites, as broadly accepted in the literature.⁴ The graphical representation for all commented models is provided in Figure S6.

In summary I do not support the publication of this manuscript on Nature Communications.

Answer:

While we respect the reviewer's opinion, we have taken their comments seriously and made significant revisions to the discussion of existing results. Additionally, we have added requested new analyses to the manuscript to address the concerns raised. We believe that these changes have substantially improved the quality and reliability of our work and are sufficient for reevaluation of the decision for publication.

Reviewer #3 (Remarks to the Author):

The work by Rzepka and co-workers aims to locate titanium atoms in the frameworks of silicalite-1 zeolite based on anomalous X-ray powder diffraction data at the Ti K-edge. The topic is of highly interest as zeolites are among the most important heterogeneous catalysts at industrial level. Furthermore, small amounts of Ti incorporated into the framework can boost their performance in oxidation reactions.

The results are very interesting and open up a new approach for the location of heteroatoms in zeolitic framework; an aspect that is highly demanded in zeolitic chemistry due to their influence in the catalytic properties that could also potentially explain their stability. There are only few aspects that I think require further clarification.

From the abstract and from the manuscript, it seems that the authors established a method for the determination of heteroatoms; however, this methodology has already been employed in zeolites. Therefore, the method is not new and the novelty comes from the analysis of Ti within the framework. I believe this statement should be better clarified.

Answer:

We appreciate the reviewer's valuable feedback. We acknowledge that the methodology we employed, anomalous X-ray powder diffraction, has been utilized by us in zeolite research previously and we would like to clarify that the novelty of our work lies rather in its specific application to precisely analyze the distribution and location of titanium within the zeolite framework. Thank you for bringing this to our attention. The following change was introduced to the abstract:

Line 23: In this study, the quantitative determination of heteroatoms within the zeolite-type framework has been enabled by exploiting anomalous X-ray powder diffraction (AXRD) and the drastic changes in titanium X-ray scattering factor at the Ti K-edge (4.96 keV).

Again, in the abstract, authors mention that the “first structure showed the large contribution from non-framework titanium in the micropores and the disorder in the remaining framework titanium. Within a unit cell of the second structure, 0.6 and 0.5 titanium atoms were found concentrated at sites T3 and T9 of MFI-type framework. Pairing of two titanium atoms can only occur at site T9 in case of highly preferred localization experienced during sample production.” They finally conclude that TS-1A contains Ti at T3 and T9; while for TS-1B the Ti is more spread over different sites (besides the extraframework Ti). The abstract and the conclusions seem contradictory, or I misunderstood something?

Answer:

The following excerpts were modified/added:

Line 26: Two representative TS-1 samples, each holding approximately 2 Ti atoms per unit cell, were examined. Half of the available titanium is primarily concentrated and distributed between sites T3 and T9 in both MFI-type frameworks, with the remaining signal dispersed among various T-sites. One of the samples exhibited a significant contribution from non-framework titanium in the micropores within the more distorted lattice. In either sample, isolated titanium atoms are notably more prevalent than dinuclear species. The latter could potentially arise only at site T9, which, however, incurs a significant energy penalty and was not detected.

Line 185: Whereas the TS-1B structure displays less distinct Bragg reflections than TS-1A (Figures S3 and S4), it also shows a titanium concentration of 1.8 primarily hosted by sites T3 (0.5(1) Ti atom) and T9 (0.4(2) Ti atom), with the remaining titanium signal spread over various T-sites. Since the reported total Ti occurrence in the sample (2.89 wt% TiO₂)¹⁴ would correspond to 2.1 Ti atoms per unit cell, the differential 0.3 titanium is non-framework.

Line 277: The framework titanium distribution in TS-1B resembles one in TS-1A, however ca. 15% titanium presence remains in non-framework locations. These two TS-1 catalysts of similar chemical composition $Si_{96-x}Ti_xO_{192}$ ($x \sim 2$) exhibit significant differences in the distribution of tetra- and octahedral titanium, despite both structures displaying elevated titanium concentration on similar T-sites within the MFI-type framework.

Another aspect that I did not follow was: on page 4, figure 1B, the authors state that both TS-1 with Ti and free are depicted. However, only TS-1 with Ti is shown.

Answer:

We appreciate the reviewer for pointing out this error. The text was modified accordingly:

Line 97: Figure 1B depicts the variations in structure factors (F_{hkl}) for TS-1 model, holding half of Ti atom in the unit cell concentrated at the T3 site when energy is switched from off- (4.600 keV) to near-resonance (4.960 keV).

About sample preparation for the measurements, in the methods section the authors describe “The powder

samples were dispersed in isopropanol and allowed to set on a 423 flat zero diffraction Si wafer allowing all orientations of crystallites within a few micrometers of the sample layer.” Would it be possible to provide SEM data of the materials to actually see if different crystallite orientations were obtained? In fact, in the manuscript it is described that vertical direction was prevented by using this preparation method.

Answer:

The SEM image of the sample subjected to the same preparation protocol was provided. We are confident that preferential orientation does not play important role in case of this study, particularly considering the spherical morphology of the crystallites. The following line was added to the main text:

Line 128: Preferred orientation of the zeolite crystallites in vertical direction was prevented by special sample preparation that is presented in the ‘Methods’ section and depicted by SEM images (Figure S7).

The following figure was added to SI:

Line 1073:

Figure S7. SEM image of sample TS-1A (left) and TS-1B (right), as prepared for the AXRD measurement.

REVIEWERS' COMMENTS

Reviewer #1 (Remarks to the Author):

The MS was resubmitted after a careful revision based on the comments and criticisms raised by the Reviewers on the first version of the paper. I'd like to thank the Authors for having seriously and scrupulously considered my suggestions and those of the other Reviewers, re-examining the existing data and performing new experimental and computational work, thus providing a more accurate interpretation, and reaching more robust conclusions.

1) Regarding the distribution of Ti in the two samples examined, the Authors agree with my reading of the original data, namely that in both the heteroatom is preferentially located in the T3 and T9 sites of the orthorhombic structure. The two samples differ in some characteristics (e.g., presence of extraframework Ti in the TS-1B, small but significant differences in the diffraction patterns) and it would be interesting to understand if these are due to different synthesis methods. I understand that, having been provided by a Company, this information is confidential; however, from my experience, it is likely that the two samples were produced under even slightly different synthesis conditions. Despite this, the experimental data agree in identifying the T3 and T9 as preferential sites in both samples and this is an aspect that should be better emphasized.

2) I thank the Authors for taking into serious consideration my suggestion to explicitly include the TPA-OH in the model used in the periodic DFT calculations. The results obtained, combined with the experimental data, strengthen the conclusion of the presence of isolated Ti sites, ruling out the recently formulated dimeric Ti sites hypothesis. This conclusion is further strengthened by the results of the Differential Pair Distribution Function analysis included in the manuscript, as suggested by Reviewer #2. To my knowledge, this is the first time that periodic DFT calculations are conducted on the "as-synthesized" TS-1 model that explicitly includes TPA-OH; the only study that explicitly included SDA was that of Hajar et al. (ref. 23, in the ms) in which TMA+ (tetramethylammonium) instead of TPA+ was used in a cluster model, but without achieving convincing results.

Some minor points:

- p. 8: the lattice parameters of the "Ti-free silicalite-1" which normally has monoclinic symmetry are reported. However, the alpha angle is not reported.
- Bibliography: Refs. 2, 3 and 9 are incomplete (journal abbreviation is missing)

In summary, although TS-1 has been the subject of a huge amount of studies, some aspects still remain to be clarified and one of these is the existence of preferential Ti sites. It is not a secondary aspect, because it represents the basis for understanding the mechanisms of oxidation reactions for which TS-1 is an excellent catalyst. The discussion on this topic is still open and the recent hypothesis of the existence of dimeric Ti sites demonstrates it. Following the Reviewers' suggestions, the Authors carried out new activities (e.g. periodic DFT calculations) and inserted new data that significantly improve the quality of the manuscript and the robustness of the results. Overall, I consider this work not only an important contribution to increasing the knowledge on TS-1 but, in some respects, it has also introduced some new methodological aspects that can be extended to the study of other zeolite catalysts. Therefore, I recommend the publication of this manuscript in Nature Communications.

Roberto Millini

Reviewer #2 (Remarks to the Author):

The revised version of the manuscript "Quantitative locating titanium in the framework of titanium silicalite-1 by exploiting anomalous X-ray powder" by Rzepka et al has now reached a high level of accuracy and it is now suitable for publication in Nature Communications.

There is however the need of a few changes that could increase the manuscript appeal to the Journal general Readership.

- It would be very interesting to modify Figure S5(b) that represents the experimentally detected differential pair distribution functions (dPDF). Indeed, in Figure S5(a) the Authors represent the PDFs up to 14 Å, while, in figure S5(b), the dPDFs are represented only up to about 6 Å. The dPDFs graphs shown in Figure S5(b) should be modified by displaying the dPDFs up to 14 Å. Such a modification could be relevant for the general discussion of the results. Indeed, the Authors explicitly state that the most probable Ti-Ti separation in TS1 is around 11-12 Å. Thus the extension of the range of the dPDFs could be of overwhelming relevance to provide further insight on the Ti distribution at the atomistic level.

- The Authors clearly show how difficult is the precise determination of the location of Ti in TS1, and actually they have changed the way adopted in the treatment of the AXRD raw data by introducing a quite new (and not standard) approach to calculate errors in the occupancy of Ti in different sites in their samples. Such difficulties are also present in DFT calculations, indeed even small differences in the calculation set-ups may lead to different results. In this respect, the Authors should quote other literature data dealing with the DFT prediction of preferential location of one or two Ti in the 12 available sites of TS1 (see e.g. J. Phys. Chem. A 2009, 113, 15006–15015, "TS1 from first principles").

- The Authors, in the revised manuscript, discuss the fallout of their structural findings on the plausible peroxy transition states structures when H₂O₂ is dosed on TS1. They mention different structures adopting nicknames like e.g. μ-η²-peroxy. Surely, specialists can easily associate such nicknames to structures, however that could be less easy for the general readership. I suggest that, in the SI, the Authors might illustrate such structures using simple graphic schemes.

- Concerning the fundamental aspect of the work, as derived from the Authors analyses, it emerges that TS1 is essentially a "single-site" catalyst, namely that the catalytic properties are due to isolated TiO₄ tetrahedra. Such an aspect should be properly discussed in the manuscript. (see e.g. Topics in Catalysis. (2006). 40 (1–4), "The advantages and future potential of single-site heterogeneous catalysts")

Ettore Fois

Reviewer #3 (Remarks to the Author):

The authors have addressed all my concerns and therefore, I recommend the work for publication.

Dear Dr. Harold Geddes,

We are pleased to resubmit our manuscript titled "Quantitative Locating Titanium in the Framework of Titanium Silicalite-1 by Exploiting Anomalous X-ray Powder Diffraction at the Ti K-edge" for publication in Nature Communications. Outlined below are our point-by-point responses to the remaining reviewers' comments.

On behalf of all authors,

Jeroen A. van Bokhoven and Silvia Bordiga

Reviewer #1 (Remarks to the Author):

The MS was resubmitted after a careful revision based on the comments and criticisms raised by the Reviewers on the first version of the paper. I'd like to thank the Authors for having seriously and scrupulously considered my suggestions and those of the other Reviewers, re-examining the existing data and performing new experimental and computational work, thus providing a more accurate interpretation, and reaching more robust conclusions.

1) Regarding the distribution of Ti in the two samples examined, the Authors agree with my reading of the original data, namely that in both the heteroatom is preferentially located in the T3 and T9 sites of the orthorhombic structure. The two samples differ in some characteristics (e.g., presence of extraframework Ti in the TS-1B, small but significant differences in the diffraction patterns) and it would be interesting to understand if these are due to different synthesis methods. I understand that, having been provided by a Company, this information is confidential; however, from my experience, it is likely that the two samples were produced under even slightly different synthesis conditions. Despite this, the experimental data agree in identifying the T3 and T9 as preferential sites in both samples and this is an aspect that should be better emphasized.

Authors response: We confirm that the two samples comes from different synthetic routes, however details are covered by a non-disclosure agreement with the company. Nonetheless, as underlined by the reviewer, the general conclusion of preferential siting of Ti across the T3/T9 crystallographic site is valid for both materials, despite the minor differences among them. We edited the text to explicitly mention the samples are produced from different synthetic routes, as it follows:

"Two industrial TS-1 samples (provided by Evonik Industries AG), originating from different synthetic routes and marked as TS-1A and TS-1B, were experimentally examined (basic characterization data available in Figure S1 and discussed therein)."

2) I thank the Authors for taking into serious consideration my suggestion to explicitly include the TPA-OH in the model used in the periodic DFT calculations. The results obtained, combined with the experimental data, strengthen the conclusion of the presence of isolated Ti sites, ruling out the recently formulated dimeric Ti sites hypothesis. This conclusion is further strengthened by the results of the Differential Pair Distribution Function analysis included in the manuscript, as suggested by Reviewer #2. To my knowledge,

this is the first time that periodic DFT calculations are conducted on the "as-synthesized" TS-1 model that explicitly includes TPA-OH; the only study that explicitly included SDA was that of Hajar et al. (ref. 23, in the ms) in which TMA+ (tetramethylammonium) instead of TPA+ was used in a cluster model, but without achieving convincing results.

Authors response: We are grateful to the reviewer for his consideration on our additional simulations and experimental investigation. We additionally explicitly referenced the attempt of Hajar et al. with TMA+ in the manuscript:

"Previous attempts of explicitly considering the role of SDA in simulation are rare and did not provide a relevant support to the interpretation of experimental data"

Some minor points:

- p. 8: the lattice parameters of the "Ti-free silicalite-1" which normally has monoclinic symmetry are reported. However, the alpha angle is not reported.

Authors response: The alpha angle is now explicitly mentioned in the caption of Figure 3

"Ti-free S-1 was refined in monoclinic P21/n11 space group with $\alpha=90.055(3)^\circ$."

- Bibliography: Refs. 2, 3 and 9 are incomplete (journal abbreviation is missing)

Authors response: The incomplete references to these three patents have been updated.

In summary, although TS-1 has been the subject of a huge amount of studies, some aspects still remain to be clarified and one of these is the existence of preferential Ti sites. It is not a secondary aspect, because it represents the basis for understanding the mechanisms of oxidation reactions for which TS-1 is an excellent catalyst. The discussion on this topic is still open and the recent hypothesis of the existence of dimeric Ti sites demonstrates it. Following the Reviewers' suggestions, the Authors carried out new activities (e.g. periodic DFT calculations) and inserted new data that significantly improve the quality of the manuscript and the robustness of the results. Overall, I consider this work not only an important contribution to increasing the knowledge on TS-1 but, in some respects, it has also introduced some new methodological aspects that can be extended to the study of other zeolite catalysts. Therefore, I recommend the publication of this manuscript in Nature Communications.

Roberto Millini

Reviewer #2 (Remarks to the Author):

The revised version of the manuscript “Quantitative locating titanium in the framework of titanium silicalite-1 by exploiting anomalous X-ray powder” by Rzepka et al has now reached a high level of accuracy and it is now suitable for publication in Nature Communications.

There is however the need of a few changes that could increase the manuscript appeal to the Journal general Readership.

- It would be very interesting to modify Figure S5(b) that represents the experimentally detected differential pair distribution functions (dPDF). Indeed, in Figure S5(a) the Authors represent the PDFs up to 14 Å, while, in figure S5(b), the dPDFs are represented only up to about 6 Å. The dPDFs graphs shown in Figure S5(b) should be modified by displaying the dPDFs up to 14 Å. Such a modification could be relevant for the general discussion of the results. Indeed, the Authors explicitly state that the most probable Ti-Ti separation in TS1 is around 11-12 Å. Thus the extension of the range of the dPDFs could be of overwhelming relevance to provide further insight on the Ti distribution at the atomistic level.

Authors response: We thank the reviewer for the suggestion. We have enlarged the range of dPDFs in Figure S5(b) accordingly. Unfortunately, most of the long-distance features match those due to defectivity in bare silicalite-1. Therefore, we cannot consistently attribute the ripples in the 11-12 Å range to Ti-Ti distances as inferred by DFT.

- The Authors clearly show how difficult is the precise determination of the location of Ti in TS1, and actually they have changed the way adopted in the treatment of the AXRD raw data by introducing a quite new (and not standard) approach to calculate errors in the occupancy of Ti in different sites in their samples. Such difficulties are also present in DFT calculations, indeed even small differences in the calculation set-ups may lead to different results. In this respect, the Authors should quote other literature data dealing with the DFT prediction of preferential location of one or two Ti in the 12 available sites of TS1 (see e.g. J. Phys. Chem. A 2009, 113, 15006–15015, “TS1 from first principles”).

Authors response: We agree with the reviewer on the criticism related to the DFT calculation setups, an issue some of us have encountered in previous literature contributions (see ref. 15). We have highlighted this issue in the text as follows:

“Known the uncertainty of DFT methods, also related to the selected computational setup, that can be in the order of 10-15 kJ/mol for the formation energy of siliceous zeolites,⁴³ it clearly emerges that few structure models account for the larger fraction of sites (6 models falls below a threshold of 15 kJ/mol).”

- The Authors, in the revised manuscript, discuss the fallout of their structural findings on the plausible peroxo transition states structures when H₂O₂ is dosed on TS1. They mention different structures adopting nicknames like e.g. μ-η²-peroxo. Surely, specialists can easily associate such nicknames to structures, however that could be less easy for the general readership. I suggest that, in the SI, the Authors might illustrate such structures using simple graphic schemes.

Authors response: We acknowledge the reviewer for the useful suggestion. We included a new Figure S8 in the SI in order to graphically render the formal nomenclature we adopted in the main text.

Figure S8. Structure of possible monomeric and dimeric Ti-peroxo species.

- Concerning the fundamental aspect of the work, as derived from the Authors analyses, it emerges that TS1 is essentially a “single-site” catalyst, namely that the catalytic properties are due to isolated TiO₄ tetrahedra. Such an aspect should be properly discussed in the manuscript. (see e.g. Topics in Catalysis. (2006). 40 (1–4), "The advantages and future potential of single-site heterogeneous catalysts")

Authors response: We acknowledge the reviewer for the interesting point he is rising. We underlined the single-site nature of TS-1, also referring to the literature suggested by the reviewer, as it follows:

“Nonetheless, provided the Ti preferentially sits at namely isolated tetrahedral sites, TS-1 can be consistently described as an heterogeneous single site catalyst, as broadly accepted in the literature.”

Ettore Fois

Reviewer #3 (Remarks to the Author):

The authors have addressed all my concerns and therefore, I recommend the work for publication.